# ITERATIVE HIERARCHICAL ATTENTION FOR ANSWERING COMPLEX QUESTIONS OVER LONG DOCUMENTS

## ABSTRACT

We propose a new model, DocHopper, that iteratively attends to different parts of long, heirarchically structured documents to answer complex questions. Similar to multi-hop question-answering (QA) systems, at each step, DocHopper uses a query $q$ to attend to information from a document, combines this "retrieved" information with $q$ to produce the next query. However, in contrast to most previous multi-hop QA systems, DocHopper is able to "retrieve" either short passages or long sections of the document, thus emulating a multi-step process of "navigating" through a long document to answer a question. To enable this novel behavior, DocHopper does not combine document information with $q$ by concatenating text to the text of $q$, but by combining a compact neural representation of $q$ with a compact neural representation of a hierarchical part of the document, which can potentially be quite large. We experiment with DocHopper on four different QA tasks that require reading long and complex documents to answer multi-hop questions, and show that DocHopper achieves state-of-the-art results on three of the datasets. Additionally, DocHopper is efficient at inference time, being 3–10 times faster than the baselines.[1]

## 1 INTRODUCTION

In this work we focus on the problem of answering complex questions over long structured documents. A long document typically contains coherent information on a certain topic, and the contents are grouped by sections or other structures. To answer complex, multi-hop questions over long documents often requires navigating through different parts of the documents to find different pieces of information relevant to a question. This navigation, in turn, requires understanding high-level information about the structure of the document. For example, to answer the question "What modules in DocHopper will be finetuned in all the experiments?", one might first turn to the section title "Model" to identify the different modules in DocHopper, and then read the "Experiments" with these modules in mind, potentially further attending to specific subsections (such as the ones titled "Implementation Details"). As in academic papers (Dasigi et al., 2021), similar tasks are common for questions about government policies (Saeidi et al., 2018) or legal documents. This type of QA tests not only the ability to understand short passages of text, but also the ability to understand the goals of the question and the structure of documents in a domain.

A common approach of solving multi-hop questions is to iteratively find evidence for one hop, and then use that evidence to update the query used in the next hop of the QA process. The update can be performed by either explicitly predicting the intermediate answers (Talmor & Berant, 2018; Sun et al., 2019) or directly appending previous evidences to the questions (Zhao et al., 2021; Qi et al., 2021; Li et al., 2020; Xiong et al., 2021). While appending retrieved evidence to a query works well on many factual QA tasks, where it is possible to answer questions with evidences that are short pieces of text, this approach is expensive if one wishes to retrieve larger pieces of text as evidences (e.g., the "experiments" section of a paper). Another disadvantage is that appending together many small fragments of text intuitively fails to capture the relationships between them, and the structure of the document from which they were extracted.

To capture high-level structural information in a document as well as detailed information from short passages, *hierarchical attention mechanisms* have been proposed, which learn neural representations

---

[1]We will open-source our code and data.

at different levels that are then mixed to make final predictions for simple questions (Wang et al., 2018; Chang et al., 2019). Hierarchical attention has also been adopted in pretrained language models, e.g. ETC (Ainslie et al., 2020), which introduced a global-local attention mechanism where embeddings of special global tokens are used to encode high-level information. Our DOCHOPPER system incorporates ETC as a document encoder. However, while ETC has previously performed well on multi-hop QA tasks like HotpotQA and WikiHop (Yang et al., 2018; Welbl et al., 2018) which require combining information from a small number of short passages, it has not been previously evaluated on tasks of the sort considered here. Our experiments show that DOCHOPPER outperforms past approaches to using ETC for multi-hop questions.

DOCHOPPER extends the existing hierarchical attention methods with a novel approach to updating queries in a multi-hop setting. DOCHOPPER iteratively attends to different parts of the document, either at fine-grained level or at higher level. This process can be viewed as either retrieving a short passage, or navigating to a part of a document. In each iteration, the query vector is updated in the embedding space rather than by re-encoding a sequence of concatenated tokens. This updating step is end-to-end differentiable and efficient. In our experiments, we also show it is effective on four different benchmarks involving complex queries over long structured documents.

In particular, we evaluate DOCHOPPER on four different tasks: conversational QA for discourse entailment reasoning, using the ShARC (Saeidi et al., 2018) benchmark[2]; factual QA with table and text, using HybridQA (Chen et al., 2020); information seeking QA on academic papers, using QASPER (Dasigi et al., 2021); and multi-hop factual QA, using HotpotQA (Yang et al., 2018). Since the outputs of the four tasks are different, additional layers or simple downstream models are appended to DOCHOPPER to get the final answers. DOCHOPPER achieves state-of-the-art results on three of datasets, outperforming the baseline models by 3%-5%. Additionally, DOCHOPPER runs 3–10 faster than the baseline models, because the neural representations of documents is pre-computed, which significantly reduces computation cost at inference time.

## 2 RELATED WORKS

Graph-based models have been widely used for answering multi-hop questions in factual QA (Min et al., 2020; Sun et al., 2018; 2019; Qiu et al., 2019; Fang et al., 2019). However, most of the graph-based models are grounded to entities, i.e., evidences (from knowledge bases or text corpus) are connected by entities in the graph. The graph construction step also heavily relies on many discrete features such as hyperlinks or entities predicted with external entity linkers. It's not clear how to apply these models to more general tasks if context is not entity-centric, such as questions about academic papers or government documents. Similar problems also exist in memory-augmented language models that achieved the state-of-the-art on many factual QA tasks (Guu et al., 2020; Lewis et al., 2021; Verga et al., 2020; Dhingra et al., 2020; Sun et al., 2021).

Alternatively, one can adopt the "retrieve and read" pipeline to answer multi-hop questions over long documents. Recent works proposed to extend the dense retrieval methods (Karpukhin et al., 2020) to multi-hop questions (Zhao et al., 2021; Qi et al., 2021; Li et al., 2020). However, such models retrieve one small piece of evidence at a time, lacking the ability of navigating between different parts of the documents to find relevant information at both higher and lower levels of the document-structure hierarchy. Another disadvantage of these iterative models is that they are not end-to-end differentiable. Updating the questions for the next hop requires re-encoding the concatenated tokens from the questions and previously retrieved evidences. It also makes the model inefficient because re-encoding tokens with large Transformer models is very expensive.

Besides question answering tasks, hierarchical attention has been successfully used in tasks such as document classification (Yang et al., 2016; Chang et al., 2019), summarization (Gidiotis & Tsoumakas, 2020; Xiao & Carenini, 2019; Zhang et al., 2019), sentiment analysis (Ruder et al., 2016), text segmentation (Koshorek et al., 2018), etc. It is worth mentioning that ETC (Ainslie et al., 2020) was also used on a key-phrase extraction task on web pages using the structured DOM tree. However, none of these models can be easily adapted to answering complex questions over long documents.

---

[2]We modify the original dataset and replace the oracle snippet with the entire web page as input. Please see section §4.1 for more details.

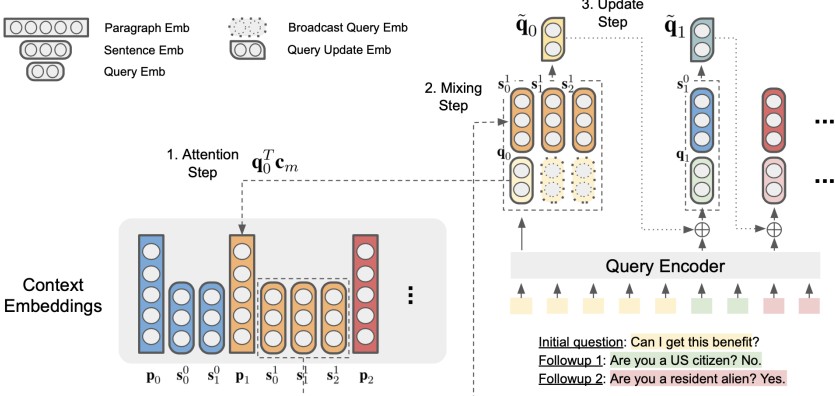

Figure 1: DocHopper Overview for a structured document consisting of sentences and paragraphs. The query encoder first computes embeddings $\mathbf{q}_0, \ldots, \mathbf{q}_n$ for a sequence of sub-questions. During the iterative attention process, DocHopper **attends** to a paragraph or a sentence from the context embedding table that contains both paragraph embeddings and sentence embeddings. Information in the attended sentence or paragraph will be **mixed** with the query vector to compute query update. If the query attends to a paragraph, e.g. the first query $\mathbf{q}_0$ in the figure, the query vector will be broadcast to the associated sentences. If the query attends to a sentence, e.g. the second query $\mathbf{q}_1$, only the sentence $\mathbf{s}_1^0$ will be used. DocHopper then **updates** the query vector $\mathbf{q}_1$ for the next round of attention. The selected sentences will be used to make final predictions.

## 3    Model

In this section, we first introduce how to compute neural representations for questions and context at different levels. Then, we present the iterative process that attends to information in the document and updates the query vector. Additional layers or simple downstream models required for different downstream tasks will be explained in the next section (§4).

### 3.1    Background

**Input** A long document usually contains multiple levels of hierarchy, e.g. sections, sub-sections, paragraphs, sentences, etc. For simplicity, we only consider two levels of hierarchy in this paper: *paragraph-level* and *sentence-level*. A *sentence* is the lowest granularity that can be retrieved, while a *paragraph* is an abstraction of a collection of sentences, which can be used to represent sections or other levels in the hierarchy, depending on the application. Formally, let $d = \{p_0, \ldots, p_{|d|}\} \in D$ be a document in the corpus $D$ that contains a sequence of paragraphs $p_i$, and let a paragraph $p_i = \{s_0^i, \ldots, s_{|p_i|}^i\}$ contain a sequence of sentences $s_0^i$. A sentence $s_j^i$ will be encoded into a fixed length vector $\mathbf{s}_j^i \in \mathbb{R}^d$.

**Pretrained ETC** We use ETC (Ainslie et al., 2020) as our query and context encoder as it is pretrained to produce both token-level and sentence-level embeddings. ETC is pretrained as a Mask Language Model (MLM). Different from BERT (Devlin et al., 2019), it employs an additional global-local attention mechanism. ETC assigns to each sentence a special *global token* that only attends to local tokens in the sentence, and its embedding is trained to summarize the information of local tokens in the sentence. A global token also attends the global tokens of other sentences in the input. ETC additionally adopts Contrastive Predictive Coding (CPC) (Oord et al., 2018) to train the embedding of global tokens to make them aware of other sentences in the context. We use the embeddings of global tokens in ETC as encodings for sentences.

Specifically, ETC takes a list of sentences $p_i = \{s_0^i, \ldots, s_{|p_i|}^i\}$ as input, and returns a list of vectors $\mathbf{s}_0^i, \ldots, \mathbf{s}_{|p_i|}^i$, where each $\mathbf{s}_j^i \in \mathbb{R}^d$ represents the embedding of a sentence $s_j^i$.

$$\mathbf{s}_0^i, \ldots, \mathbf{s}_{|p_i|}^i = \text{ETC}(\{s_0^i, \ldots, s_{|p_i|}^i\}) \ \in \mathbb{R}^{|p_i| \times d}$$

## 3.2 QUERY EMBEDDINGS

We discuss the strategy of computing query vectors for two types of input, conversational QA and multi-hop QA. We use ETC (Ainslie et al., 2020) as our query encoder for the reasons discussed above. Other pretrained language models for similar purposes should work as well.

**Conversational QA** A conversational question contains multiple rounds of interaction between the machine and human. As the conversation goes on, the topic of the conversation will shift, but questions asked in the next turn are related to the history of the conversation. We consider this multi-turn process as a multi-hop QA task. In our specific task, the conversation starts with an initial question $q_0$. The model will answer the question if there's enough information. Otherwise, the model will ask a followup question $f_1$ and expect the user's answer $a_1$. The model will keep asking followup questions for a few more iterations. The task is to predict the answer if there is enough information after reading the full conversation history, or mark the question as not answerable.

We denote $a_i$ as the answer to the followup question $f_i$ received from the users at the $i$'th iteration. We consider the full conversation history as the question $p_q = \{q_0, (f_1, a_1), \ldots, (f_n, a_n)\}$ with the initial question denoted $q_0$ and the $i$-th followup question-answer pair denoted $(f_i, a_i)$. The answer $a_i$ to the followup question $f_i$ is concatenated to the end of the questions and represented as a single sentence. We call $p_q$ a *question paragraph* as it contains a sequence of question-answer pairs. We use ETC to compute the query embeddings $\mathbf{q}_i$, one for each sentence in the query paragraph $p_q$. The query embeddings $\mathbf{q}_i$ will be used to perform iterative attention over the document.

$$\mathbf{q}_0, \ldots, \mathbf{q}_n = \text{ETC}_q(\{q_0, (f_1, a_1), \ldots, (f_n, a_n)\}) \tag{1}$$

**Multi-hop QA** A multi-hop question, e.g. "*Which gulf is north of the Somalian city with 550,000 residents*", requires the model to first find the answer of the first "hop" of the question, and then perform a second round to find the final answer. Assume for now there are exactly two hops. Different from conversational QA, questions in multi-hop QA do not have a clear split between the two hops of questions, making it impossible to explicitly split a question into two sub-questions. Instead, we add a dummy question $q_{\text{null}}$ to the question paragraph $q_p = \{q_0, q_{\text{null}}\}$. The global-to-local attention mask of ETC is modified to allow the global token of the dummy question to attend to tokens in the question $q_0$. With this modification, the query embeddings $\mathbf{q}_0$ and $\mathbf{q}_1$ for $q_0$ and $q_{\text{null}}$ can to attend any part of the question, but each can also attend to different parts of the question. One could append additional dummy questions to the question paragraph $p_q$ if the number of hops is larger. Thus we let

$$\mathbf{q}_0, \mathbf{q}_1 = \text{ETC}_q(\{q_0, q_{\text{null}}\}) \tag{2}$$

Note that we assume the true number of hops is known in multi-hop QA tasks, which is two or four for the experiments in this paper. However, one can train models to decide when to stop if questions have various numbers of hops. We leave this as a topic for future work.

## 3.3 NEURAL REPRESENTATIONS OF PARAGRAPHS

We compute embeddings $\mathbf{s}_j^i$ for the sentence $s_j^i$ with the context encoder. We use ETC as our context encoder, but similar pretrained language models will also work. Recall that a paragraph $p_i = \{s_0^i, \ldots, s_{|p_i|}^i\}$ contains a sequence of sentences $s_j^i$. The sentence embeddings $\mathbf{s}_j^i$ in the paragraph $p_i$ are simply computed by applying ETC on the paragraph input.

$$\mathbf{s}_0^i, \ldots, \mathbf{s}_{|p_i|}^i = \text{ETC}_c(\{s_0^i, \ldots, s_{|p_i|}^i\})$$

The paragraph embeddings are directly derived from sentence embeddings $\mathbf{s}_j^i$ and dependent on the queries $\mathbf{q}_t$, the embedding of the $t$'th hop of the question. The paragraph embedding $\mathbf{p}_i$ is a weighted sum of sentence embeddings $\mathbf{s}_j^i$ in the paragraph $p_i$, where $\alpha_j$ is the attention weights of the query vector $\mathbf{q}_t$ to the sentence embedding $\mathbf{s}_j^i$. The paragraph embeddings $\mathbf{p}_i$ are thus dependent on the query, but do not require jointly encoding tokens from queries and context, as in many BERT-style reading comprehension models. Computing paragraph embeddings with Eq.3 is hence very efficient.

$$\mathbf{p}_i = \sum_j \alpha_j \, \mathbf{s}_j^i, \quad \alpha_j = \text{softmax}(\mathbf{q}_t^T \mathbf{s}_j^i) \tag{3}$$

### 3.4 ITERATIVE ATTENTION

We put the sentence embeddings and paragraph embeddings of document $d$ into a combined *context embedding table*, so the model has the flexibility to decide which sentence or paragraph to attend to. Different update rules will be applied according to whether sentences or paragraphs are attended to.

To construct the embedding table, we iterate through all paragraphs in a document and apply the ETC encoder on each paragraph to compute the paragraph and sentences embeddings. Sentence and paragraph embeddings from all paragraphs are then merged together to form the combined embedding table. We denote the combined embedding table for document $d$ as $\mathbf{C}_d = \{\mathbf{p}_0, \mathbf{s}_0^0, \dots, \mathbf{s}_{|p_0|}^0, \mathbf{p}_1, \mathbf{s}_0^1, \dots, \mathbf{s}_{|p_1|}^1, \dots\}$. Let $\mathbf{c}_m$ be the embedding of the $m$'th entry from $\mathbf{C}_d$; we emphasize that $\mathbf{c}_m$ can represent either a sentence or a paragraph embedding.

**Attention Step** At each iteration, DOCHOPPER computes the inner product scores between the query vector $\mathbf{q}_t$ and embeddings $\mathbf{c}_m$ in $\mathbf{C}_d$, and returns the entry $\hat{c}$ with the largest score, which is usually referred as hard attention. (As we will see $\hat{c}$ is not directly used for computation, but it is helpful in explaining the attention method).

$$\hat{c} = \mathrm{argmax}_{c_m}(\mathbf{q}_t^T \mathbf{c}_m)$$

**Mixing Step** DOCHOPPER then mixes the embedding of the attended entry $\hat{c}$ with the query vector $\mathbf{q}_t$ to find the missing information. Since the combined embedding table $\mathbf{C}_d$ contains both sentence and paragraph embeddings, the selected entry $\hat{c}$ can represent either a sentence or a paragraph. The two cases are separately considered. If $\hat{c}$ is a sentence, i.e. $\hat{c} = s_j^i$, DOCHOPPER computes the mixed embeddings as

$$\tilde{\mathbf{q}}_t = \mathbf{W}_q^T [\mathbf{q}_t; \mathbf{s}_j^i] \tag{4}$$

where $[\mathbf{q}_t; \mathbf{s}_j^i]$ is the concatenation of two vectors $\mathbf{q}_t$ and $\mathbf{s}_j^i$. Assume $\hat{c}$ is a paragraph, i.e., $\hat{c} = p_i$, DOCHOPPER first looks up sentences in $p_i$, i.e. the list $\{s_0^i, \dots, s_{|p_i|}^i\}$. The following process is then used to compute the mixed embedding $\tilde{\mathbf{q}}_t$: (1) DOCHOPPER computes the attention weights of the query vector $\mathbf{q}_t$ to the embeddings of associated sentences $\{s_0^i, \dots, s_{|p_i|}^i\}$ that measures the relevance scores between the query vector and the sentences. This attention weight is the same as the weight $\alpha_j$ in Eq.3 that is used to compute the paragraph embeddings, so we reuse $\alpha_j$ in this equation. In the implementation, we also re-use the value of $\alpha_j$ if it has been computed for the query-dependent paragraph embeddings. (2) The query vector $\mathbf{q}_t$ is compared with every sentences in paragraph to extract missing information. $\mathbf{q}_t$ is broadcast to the sentence embedding $\mathbf{s}_j^i$ with the weight $\alpha_j$, i.e. the query vector is more important if more relevant. The comparison is performed as a linear projection of the concatenated query and sentence embeddings, $\mathbf{q}_t$ and $\mathbf{s}_j^i$.

$$\mathbf{k}_j = \mathbf{W}_q^T [\alpha_j \, \mathbf{q}_t; \mathbf{s}_j^i] \tag{5}$$

Then (3) the concatenated vectors $\mathbf{k}_j$ are summed with the weight $\beta_j$, where $\beta_j$ is the attention weight of a learned vector $\mathbf{v}$ to the concatenated vector $\mathbf{k}_j$. The learned vector $\mathbf{v}$ coordinates the importance of sentences from the attended paragraph after comparing them with the query vector and decides what information to pass to the next step of retrieval.

$$\tilde{\mathbf{q}}_t = \sum_j \beta_j \, \mathbf{k}_j, \quad \beta_j = \mathrm{softmax}(\mathbf{v}^T \mathbf{k}_j) \tag{6}$$

It is not hard to see that computing the mixed embedding in Eq. 6 for the case that a paragraph is retrieved is essentially the same as in Eq. 4 if the retrieved paragraph $p_i$ only contains one sentence, i.e. $\alpha_j = 1$ and $\beta_j = 1$ if $|p_i| = 1$; hence the same logic can be used regardless of whether $\hat{c}$ is a sentence or a paragraph.

**Update Step** The mixed embedding $\tilde{\mathbf{q}}_t$ is then used to update the query vector $\mathbf{q}_{t+1}$ for the next step. Intuitively, $\tilde{\mathbf{q}}_t$ is the residual from the previous step. Adding the residual embedding encourages the model to attend to information that is not fully satisfied from previous steps.

$$\mathbf{q}_{t+1} \leftarrow \mathbf{q}_{t+1} + \tilde{\mathbf{q}}_t \tag{7}$$

**Loss Function** Attention is supervised if (distantly) supervised labels are available in the dataset. $\mathbf{q}_t^T \mathbf{c}_m$ is the inner product score between the query vector $\mathbf{q}_t$ and a context embedding $\mathbf{c}_m$. $\mathbb{I}_{c_m}$ is an indicator function that equals to 1 iff the label of $c_m$ is positive.

$$l_t = \text{cross\_entropy}(\text{softmax}(\mathbf{q}_t^T \mathbf{c}_m), \mathbb{I}_{c_m})$$

The loss function is computed at the final step, and possibly at intermediate steps if labels are available. Supervision labels are sometimes distantly constructed. For example, in the extractive QA task, a positive candidate is the sentence or paragraph that contains the answer span (see §4).

## 3.5 RUNTIME EFFICIENCY

DOCHOPPER is very efficient at runtime thanks to the precomputed context embeddings (§3.3) at inference time. Different from previous reading comprehension models that jointly encode questions and context (Beltagy et al., 2020; Ainslie et al., 2020), DOCHOPPER encode question embeddings and context embeddings independently. At inference time, DOCHOPPER lets questions directly attend to the precomputed context embeddings, significantly reducing the computation cost compared to cross-attention models that jointly encode questions and context.

## 4 EXPERIMENTS

We evaluate DOCHOPPER on four different datasets: ShARC (Saeidi et al., 2018), HybridQA (Chen et al., 2020), QASPER (Dasigi et al., 2021), and HotpotQA (Yang et al., 2018).[3] Since the downstream tasks of the four datasets are different, we apply an additional layer for ShARC, and a Transformer-based extractive QA model for HybridQA, QASPER and HotpotQA to extract the final answers. We will discuss the setup for each case separately.

### 4.1 EXTRACTIVE QA: HYBRIDQA AND QASPER

**Dataset** HybridQA is a dataset that requires jointly using information from tables and hyperlinked text from cells to find the answers. Please see the Appendix for more information. This dataset requires the model to first locate the correct row in the table, and then find the answer from cells (or their hyperlinked text). To apply DOCHOPPER on the HybridQA dataset, we first convert a table with hyperlinked text into a long document. Each row in the table is considered a paragraph by concatenating the column header, cell text, and hyperlinked text if any. The average length of the documents is 9345.5 tokens.

**Dataset** QASPER (Dasigi et al., 2021) is a QA dataset constructed from NLP papers. The dataset contains a mixture of extractive, abstractive, and yes/no questions. We experiment with the subset of extractive questions (51.8% of the datasets) in this paper. Some questions in the dataset are answerable with a single-hop of retrieval. However, as described in the original paper, 55.5% of the questions have multi-paragraph evidence, and thus aggregating multiple pieces of information should improve the accuracy. Answers in the QASPER dataset have an average of 14.4 tokens. We treat each subsection as a paragraph and prepend the section title and subsection title to the beginning of the subsection.

**Implementation Details** Question embeddings $\mathbf{q}_0, \mathbf{q}_1$ are initialized from Eq. 2 and updated as in Eq. 7. For the two datasets, we perform 2-hop attention: the first hop is trained to attend to a paragraph, and the second hop to attend to a sentence. Note that we do not require that sentences attended to in the second hop must come from the previously attended paragraphs. The attention scores of paragraphs and sentences are linearly combined to find the best sentence that contains the answer, which will then be read by a BERT-based reader to extract the final answer. The final attention score of a sentence $s_j^i \in p_i$ is

$$\text{score}(s_j^i) = \mathbf{q}_1^T \mathbf{s}_j^i + \lambda_1 \cdot \mathbf{q}_0^T \mathbf{p}_i + \lambda_2 \cdot \text{sparse}(q_0, p_i) \tag{8}$$

---

[3]We modify the original ShARC and HotpotQA datasets to evaluate them in the long-document setting. Please see the dataset descriptions for more details.

| | HybridQA | | QASPER (Extractive) | | |
| | Dev | Test | Dev | Test | runtime |
|---|---|---|---|---|---|
| Retrieval + ETC | 37.0 / 43.5 | 34.1 / 40.3 | 8.3 / 18.7 | 9.6 / 19.1 | 8.3/s |
| Sequential (ETC) | 39.4 / 44.8 | 37.0 / 43.0 | 11.2 / 24.6 | 12.4 / 27.0 | 0.6±0.1/s |
| Hybrider | 44.0 / 50.7 | 43.8 / 50.6 | – / – | – / – | 5.1/s |
| LED | – / – | – / – | – / 26.1 | – / 31.0 | 0.5/s |
| DOCHOPPER | 47.7 / 55.0 | 46.3 / 53.3 | **14.0 / 29.6** | **19.5 / 36.4** | 74.6/s |
| (w/o sparse) | 44.4 / 51.2 | – / – | (14.0 / 29.6) | – / – | – |
| (w/o query update) | 44.2 / 50.9 | – / – | 12.2 / 28.0 | – / – | – |
| (sentence-only) | 36.7 / 43.7 | – / – | 11.4 / 27.2 | – / – | – |
| (single-hop) | 27.8 / 34.1 | – / – | 11.8 / 27.3 | – / – | – |
| (w/ cell) | 53.1 / 61.4 | – / – | – / – | – / – | – |
| MATE * | **63.4 / 71.0** | **62.8 / 70.2** | – / – | – / – | – |

Table 1: EM/F1 performance on HybridQA and QASPER. Runtime is measured by reruning their open-sourced codes (failed to rerun MATE). Numbers for QASPER are reported on the subset of extractive questions. * See Results and Analysis for discussion on MATE's performance.

where $\mathbf{q}_1^T \mathbf{s}_j^i$ and $\mathbf{q}_0^T \mathbf{p}_i$ are the attention scores at sentence and paragraph levels, and $\text{sparse}(q_0, p_i)$ is the similarity score using sparse features.[4] $\lambda_1$ and $\lambda_2$ are hyper-parameters tuned on dev data. We set $\lambda_1 = 1.5$ and $\lambda_2 = 3.0$ for HybridQA, and $\lambda_1 = 0.5$ and $\lambda_2 = 0.0$ for QASPER.

**Baselines** We compare DOCHOPPER with the previous state-of-the-art models, MATE (Eisenschlos et al., 2021) and LED (Dasigi et al., 2021), and several other competitive baselines to show the efficacy of DOCHOPPER. MATE (Eisenschlos et al., 2021) is a pretrained Transformer model with sparse attention between cells that is specifically designed for tabular data. HYBRIDER (Chen et al., 2020) is a pipeline system that (1) links cells, (2) reranks linked cells, (3) hops from one cell to another, and (4) reads text to find answers. All four stages are trained separately. LED is an encoder-decoder model that builds on Longformer (Beltagy et al., 2020). We also experiment with directly reading the documents with a Transformer-based reader ETC (Ainslie et al., 2020): though it can't fit the entire document into its input, it is still one of the best models for reading long sequences (up to 4096 tokens). To handle longer documents, we adopt the sequential reading strategy: the model reads the document paragraph by paragraph, and picks the most confident prediction as the answer. We also report the numbers of a "retrieve and read" pipeline with a BM25 retriever and a finetuned ETC reader. The numbers are shown in Table 1. Runtime is measured as examples per second with a batch size of 1.

**Results and Analysis** On QASPER, DOCHOPPER outperforms the previous state-of-the-art models by 3-5%, and runs more than 10 times faster. DOCHOPPER performs worse than MATE (Eisenschlos et al., 2021) on HybridQA. This is due to three reasons. (1) MATE is specifically designed and pretrained to understand tabular data, while DOCHOPPER is applied to general documents. (2) To convert tables to DOCHOPPER's input, DOCHOPPER serializes the tables by rows and thus loses information from other table structures, e.g. cells and columns, that are commonly used to understand tabular data. In an ablated experiment with the assumption that sentences are grouped by cells, we let DOCHOPPER return cells that contain the selected sentences and pass the cells (instead of single sentences) to the underlying extractive QA model (labeled w/ cell). This improves the performance by 5.4 points. (3) MATE restricts the length of text in cells to 5 sentences and enforces it by retrieving top-$k$ sentences from the hyperlinked text. It also restricts the total length of tables to 2048 tokens. DOCHOPPER does not put any restrictions on the length of cell or the length of tables, and can thus be easily applied to more general tasks.

We additionally performed more ablated experiments with DOCHOPPER. The query update (see the row w/o query update) in Eq. 7 is also important, causing 3.5% and 1.8% difference in performance on both datasets. We also ablated the model by using one step of attention to select the most relevant sentence from the document (single-hop), and note again that performance drops noticeably. Adding one more step of attention, while only attending to sentences (sentence-only in the table), leads to

---

[4]Similar to the baseline model (Chen et al., 2020), we use paragraph-level sparse features to improve accuracy. In HybridQA, $\text{sparse}(q_0, p_i)$ computes the length of longest common substrings in the question $q_0$ and the paragraph $p_i$. Sparse features are only used at the end of retrieval, not at any intermediate steps.

| | HotpotQA-Long | |
| | Answer | Support |
| --- | --- | --- |
| IRRR (direct) | 53.6 / 64.8 | 44.7 / 73.2 |
| IRRR (rerank) | 62.1 / 75.6 | 54.5 / 81.3 |
| DOCHOPPER (direct) | 57.4 / 69.5 | 45.1 / 73.8 |
| DOCHOPPER (rerank) | **66.5 / 79.7** | **61.4 / 85.9** |
| ETC (distractor) | – / 81.7 | – / 89.4 |
| HGN (distractor) | – / 83.4 | – / 89.2 |

Table 2: EM/F1 performance on answer and supporting evidence on HotpotQA (dev set) with full Wikipedia pages as context. HGN Fang et al. (2019) and ETC Fang et al. (2019) are evaluated on the distractor setting.

| | ShARC-Long | | |
| | Easy | Strict | runtime |
| --- | --- | --- | --- |
| ETC | 61.1 | – | 11.2/s |
| Retrieval + DISCERN | 65.2 | 50.7 | 42.0/s |
| Sequential (DISCERN) | 63.7 | 54.2 | 8.8/s |
| DOCHOPPER | 72.3 | **60.2** | 164.3/s |
| (w/o query update) | **72.4** | 59.9 | – |
| (sentence only) | 62.2 | 43.2 | – |
| (single-hop) | 68.0 | 52.7 | – |

Table 3: Classification accuracy on ShARC-Long dataset. The *Easy* setting only checks the predicted labels, while the *Strict* setting additionally checks if all required evidences are retrieved. DISCERN is run with their open-sourced codes.

some improvement, but is still worse than attending at both paragraph and sentence levels. The performance of sentence selection (with ablated numbers) is presented in the Appendix.

## 4.2 HOTPOTQA-LONG

**Dataset** HotpotQA requires multi-hop reasoning to answer two types of questions, *bridge* and *comparison*. The original dataset provides the first paragraphs of 10 Wikipedia pages as its context. To test the model's ability to extract information from a longer context, we use first 2048 tokens of the 10 Wikipedia pages, and concatenate the 10 pages into a single document. This increases the average length of context from 897 to 9970 tokens. We call this variant of the dataset *HotpotQA-Long*.

**Implementation Details** The experiment setup for HotpotQA-Long is similar to HybridQA and QASPER. Attention for this dataset is supervised, using the information about which evidences support each answer provided in the dataset. Here, we refer to a Wikipedia page (with a maximum of 2048 tokens) as a *paragraph*. The iterative attention is performed similarly to §4.1, but repeated for 4 hops: paragraph, sentence, paragraph, and sentence. The first two hops are supervised by the first supporting evidence, and the last two hops are supervised by the second supporting evidence. The final score for each evidence is computed as in Eq. 8 with $\lambda_1 = 0.7$ for the first evidence and $\lambda_1 = 0.0$ for the second, and $\lambda_2 = 0$ for both. As suggested by Xiong et al. (2021), reranking can improve the overall performance, so we take the top 4 sentences for each step and rerank the 16 combinations.[5]

**Baselines** We compare DOCHOPPER with IRRR (Xiong et al., 2021) which is an multi-hop retrieval model that iteratively retrieves a small piece of evidence and re-encodes query vectors at each step by concatenating tokens from the retrieved passages at the previous step.[6] We compare to IRRR both with or without the reranking step. Since questions in HotpotQA require combining evidences from multiple paragraphs, it's not clear how to run the sequential reading baseline on HotpotQA-Long, but as a reference, we show the performance of ETC and HGN in the distractor setting (with the original context, which is 10x smaller) in Table 2.

**Results** DOCHOPPER outperforms the baseline models by 4-6 points on the accuracy of both answer and supporting evidence. More surprisingly, the performance of DOCHOPPER in the long document setting is only ∼4 points lower than the state-of-the-art on the distractor setting, even though the context is more than 10 times longer. Please see the Appendix for ablated experiments and analysis.

## 4.3 CONVERSATIONAL QA: SHARC-LONG

**Dataset** ShARC Saeidi et al. (2018) is a conversational QA dataset for discourse entailment reasoning. Each example consists of a document that describes a government policy and a conversation history about the document between the machine and the user. The conversation starts with an initial questions asked by the user and follows by a sequence of clarification questions and answers collected from the interaction between the machine and the user. The model takes the conversation

---

[5]Please refer to Xiong et al. (2021) for more details on the reranking step.

[6]IRRR is the best model (ranked #9 on leaderboard) on HotpotQA (fullwiki) with open-sourced codes.

history as input and predicts the answer to the initial question. An answer has one of the four labels: "Yes", "No", "Irrelevant", or "Inquire". "Irrelevant" means the question is not related to the provided context and "Inquire" means there's not enough information to answer the question. Similar to HotpotQA-Long, we expand the original context to test the long document setting. The expanded context contains 737.1 tokens on average, 13.5 times longer than the original context. Please see the Appendix for more details and examples.

**Implementation Details** The query embeddings $\mathbf{q}_0, \ldots, \mathbf{q}_n$ are initialized from Eq. 1 where $n$ is the number of followup questions in the conversation. The query vector $\mathbf{q}_t$ at the step $t$ will be updated with the residuals following the update rule in Eq.7. The iterative attention process is distantly supervised at intermediate steps. See the Appendix for information on distant supervision.

We add a simple classification layer on DOCHOPPER to make the final prediction. As input to this layer, we reuse the concatenated vector $\mathbf{k}_j$ in Eq. 5. Let $\mathbf{k}_j^{(t)}$ be the concatenated vector of the sentence attended at the $t$'th retrieval step, either a sentence directly attended or associated from a attended paragraph. Concatenated embeddings at all attention steps $\{\mathbf{k}_0^{(0)}, \ldots, \mathbf{k}_*^{(0)}, \ldots, \mathbf{k}_0^{(n)}, \ldots, \mathbf{k}_*^{(n)}\}$ are linearly combined into one vector $\tilde{\mathbf{k}}$ that will be used to make the final prediction, using weights $\gamma_j^{(t)}$ computed across the attended sentences from all steps $t$. $\mathbf{m} \in \mathbb{R}^4$ holds the logits of the 4 classes that is used to compute the softmax cross entropy with the one-hot encoding of the positive class labels.

$$\tilde{\mathbf{k}} = \sum_t \sum_j \gamma_j^{(t)} \mathbf{k}_j^{(t)}, \quad \gamma_j^{(t)} = \text{softmax}(\mathbf{u}^T \mathbf{k}_j^{(t)}), \quad \mathbf{m} = \mathbf{W}_c^T \tilde{\mathbf{k}} \in \mathbb{R}^4$$

**Baselines** We compare our model with ETC and a "retrieve and read" baseline. In using ETC, we concatenate the full context and the conversation into a single input, and prepend a global [CLS] token. The embedding of the global [CLS] token will be used to make the final prediction. For the "retrieve and read" pipeline, we adopt the previous state-of-the-art model DISCERN as the reader, and pair it with a learned retriever.[7] We also run a sequential reading baseline with DISCERN where documents are chunked every 128 tokens with a stride of 32 tokens and then read with DISCERN, predicting the class with the highest probability among all chunked inputs.[8] Runtime is measured as examples per second.

**Results** The evaluation is performed in two settings: *Easy* and *Strict*. The *Easy* setting only evaluates the accuracy of classification. In the *Strict* setting, we additionally require that all evidences (provided in the original dataset) are retrieved. We report the (micro) accuracy as the evaluation metric. DOCHOPPER outperforms all baseline models by more than 7 points in both easy and strict settings, while being more than 3 times faster than all baseline models. We performed the previous ablated experiments and observed similar changes in performance. Please see the Appendix for more numbers on evidence selection.

## 5   CONCLUSION

We consider on the problem of answering complex questions over long structured documents. Like multi-hop open QA tasks, this problem requires not only conventional "machine reading" abilities, but the ability to retrieve relevant information and refine queries based on retrieved information. Additionally, it requires the ability to navigate through a document, by understanding the relationship between sections of the document and parts of the question. In our framework, navigation is modeling similarly to retrieval in multi-hop models: the model attends to a document section, and uses a compact neural encoding of the section to update the query. Unlike most prior multi-hop QA models, however, queries are updated in embedding space, rather than by appending to a discrete representation of question text. This approach is end-to-end differentiable and very fast. Experiments also demonstrate that this use of hierarchical attention can significantly improve the performance on QA tasks: in fact, the DOCHOPPER model achieves the start-of-the-art results on four challenging QA datasets, outperforming the baseline models by 3–5%, while also being 3–10 times faster.

---

[7]The retriever is a ablated DOCHOPPER model that attends at the paragraph level only.

[8]The model is often extremely confident on the "Irrelevant" class because most chunked inputs are obviously irrelevant. We tuned a hyper-parameter $\epsilon = 0.99$ so the model predicts "Irrelevant" if the probabilities of all chunked inputs are larger than $\epsilon$.

## 6 ETHICS STATEMENT

Pretrained language models (LM) are effective in many tasks but are criticized due to their high training cost and environmental concerns. Recently, more attention has been brought into development efficient models. The DOCHOPPER model proposed in this paper follows research in this direction. DOCHOPPER exploits the existing language models released by other research institutes. Without any additional pretraining, DOCHOPPER achieved competitive or state-of-the-art results in several challenging QA datasets. All experiments in this paper were conducted with a single GPU.

Besides being efficient in training, DOCHOPPER is also extremely efficient during inference. Experiments show that DOCHOPPER runs 3-10 times faster compared to existing QA models and thus could be deployed using limited computation resources.

## 7 REPRODUCIBILITY STATEMENT

Experiments in this paper are conducted on publicly available datasets. Codes and processed data will be open-sourced upon the acceptance of this paper.

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

# A    APPENDIX

## A.1    DATASET DETAILS

**HybridQA** (Chen et al., 2020) is a dataset that requires jointly using information from tables and text hyperlinked from table cells to find the answers of multi-hop questions. A row in the table describes attributes of an instance, for example, a person or an event. Attributes are organized by columns. For example, the table of Medalist of Sweden in 1932,[9] contains a row "[Medal:] *Gold*; [Name:] *Rudolf Svensson*; [Sport:] *Wrestling (Greco-Roman)*; [Event:] *Men's Heavyweight*". Text in the square brackets are the headers of the table. The medal winner "*Rudolf Svensson*" and the event "*Wrestling (Greco-Roman)*" are hyperlinked to the first paragraph of their Wikipedia pages. A question asks "*What was the nickname of the gold medal winner in the men's heavyweight greco-roman wrestling event of the 1932 Summer Olympics?*" requires the model to first locate the correct row in the table, and find the answer from other cells in the row or their hyperlinked text.

To apply our model on the HybridQA dataset, we first convert a table with hyperlinked text into a long document. Each row in the table is considered a paragraph by concatenating the column header, cell text, and hyperlinked text if any. The column name and cell text are each treated as one sentence. Hyperlinked text is also split into sentences. In the example above, the row becomes "*Medal. Gold. Name. Rudolf Svensson. Johan Rudolf Svensson (27 March 1899 – 4 December 1978) was a Swedish wrestler. He competed ...*". The average length of the documents is 9345.5.

**QASPER** (Dasigi et al., 2021) is a QA dataset constructed from NLP papers. They hired graduate students to read the papers and ask questions. A different group of students are hired to answer the questions. For example, a question asks "What are the baseline models used in this paper?". The answers are {"*BERT*", "*RoBERTa*"}. The dataset contains a mixture of extractive, abstractive, and yes/no questions. We focus on the subset of extractive questions (51.8% of the datasets) in this paper. Some questions in the dataset are answerable with a single-hop. However, as suggested in the original paper, 55.5% of the questions have multi-paragraph evidence, and thus aggregating multiple pieces of information should improve the accuracy. Answers in the QASPER dataset are longer, with an average of 14.4 tokens. We treat each subsection as a paragraph and prepend the section title and subsection title to the beginning of the subsection.

**ShARC** Saeidi et al. (2018) is a conversational QA dataset for discourse entailment reasoning. Questions in ShARC are about government policy crawled from government websites. Users engage with a machine to check if they qualify for some benefits. A question in the dataset starts with a

---

[9]https://en.wikipedia.org/wiki/Sweden_at_the_1932_Summer_Olympics

|  | HybridQA | QASPER (Extractive) |
|---|---|---|
| DocHopper | **56.5** | **39.1** |
| (w/o sparse) | 53.3 | (39.1) |
| (w/o query update) | 51.8 | 37.2 |
| (sentence-only) | 46.4 | 36.1 |
| (single-hop) | 34.2 | 36.8 |

Table 4: Hits@1 accuracy on selecting sentences that actually contains the answer (on dev set).

initial question, e.g. *"Can I get standard deduction for my federal tax return?"*, with a user scenario, e.g. *"I lived in the US for 5 years with a student visa"*, and a few followup questions and answers through the interaction between the machine and users, e.g. "Bot: *Are you a resident alien for tax purpose? User: No*". The model reviews the conversation and predicts one of the three labels: "Yes", "No", or "Irrelevant". If the model think there's not enough information to make the prediction, it should predict a fourth label "Inquire".

Besides the conversation, each example in the ShARC dataset provides a snippet that the conversation is originated from. A snippet is a short paragraph that the conversation is created from, e.g. *"Certain taxpayers aren't entitled to the standard deduction: (1) A married individual filing as married... (2) An individual ..."*. Since the snippets are usually short, with an average of 54.7 tokens, previous models, e.g. DISCERN Gao et al. (2020), concatenate the snippet and the conversation, and jointly encode them with Transformer-based models, e.g. BERT or RoBERTa. Here we consider instead a more challeging long-document setting, in which the snippet is not known, and the model must also locate the snippet from the document. We crawl the web pages with the provided URL. The pages contain 737.1 tokens on average, 13.5 times longer than the original snippets, and the longest page contains 3927 tokens. We name this new variant ShARC-Long.

## A.2 Implementation Details for ShARC-Long

**Changes to Context Representations** Instead of computing the paragraph embeddings as a weighted sum of sentence embeddings, we directly obtain the paragraph embeddings from ETC output for this dataset. Recall that a paragraph $p_i = \{s_0^i, \ldots, s_{|p_i|}^i\}$ contains a sequence of sentences $s_j^i$. We prepend a dummy sentence $s_{\text{null}}$ to the beginning of the paragraph, and again, we modify the global-to-local attention mask to allow the global token of the dummy sentence to attend to all tokens in the paragraph $p_i$. Let $\mathbf{p}_i \in \mathbb{R}^d$ be the embedding of paragraph $p_i$. The embeddings for a paragraph and its contained sentences are:

$$\mathbf{p}_i, \mathbf{s}_0^i, \ldots, \mathbf{s}_{|p_i|}^i = \text{ETC}(\{s_{\text{null}}, s_0^i, \ldots, s_{|p_i|}^i\})$$

**Distant Supervision** The iterative attention process is distantly supervised with supervision at intermediate steps. At each step, the model is trained to attend to both the correct paragraph and the correct sentences if they exists. Since the embedding table $\mathbf{C}_d$ consists of both paragraph and sentence embeddings, we only need to compute the attention scores once at each step, but consider both the correct paragraph and the correct sentence as positive. The positive paragraph is one of the paragraphs from the crawled web page with the highest BLEU score.[10] We notice that some web pages at the provided URLs have been changed significantly, so the snippets provided in the datasets may not exist any more, hence we discard the associated data if the highest BLEU scores of the paragraphs is less than 0.7. We follow the heuristics used by baseline models (Gao et al., 2020) to get positive sentence candidates by finding the sentence with the minimum edit distance.

## A.3 Additional Results

We report the performance of eventually selecting the correct evidences in Table 4, 5, and 6.

**Comments on HotpotQA-Long** We also observe that ablated experiment on evidence selection (w/o query update) is only 7.8 points lower than the full model. To understand the underlying

---

[10] We drop the brevity penalty term in BLEU score.

|  | HotpotQA-Long |
|---|---|
| IRRR | 56.8 |
| DOCHOPPER | 64.7 |
| (w/o query update) | 56.5 |
| (sentence-only) | 61.8 |
| (single-hop) | 37.4 |

Table 5: Accuracy of correctly predicting supporting facts for both hops on HotpotQA-Long (without reranking).

|  | ShARC-Long |
|---|---|
| DOCHOPPER | 82.2 |
| (w/o query update) | 81.8 |
| (sentence-only) | 63.0 |
| (single-hop) | 72.4 |

Table 6: Accuracy of selecting all required evidences on ShARC-Long.

reason, we train the model to perform a one-step attention only for supporting facts of the second hop (for bridge questions). The accuracy is 71.7, only 3.6 points lower than the accuracy of the full multi-hop process. This is likely due to the high surface form overlap between the questions and their context.

