# OpenReview forum: "Iterative Hierarchical Attention for Answering Complex Questions over Long Documents"
_ICLR.cc/2022/Conference — ICLR 2022 Submitted_

### Official Review · Reviewer_XkRC · 2021-11-02

**Correctness:** 2
**Technical Novelty And Significance:** 2
**Empirical Novelty And Significance:** 2
**Recommendation:** 3
**Confidence:** 4

**Main Review:**

Strengths:

1. This paper proposes a machine reading comprehension model, which is computationally efficient and also can handle long-document in multiple granularities.
2. The computational efficiency of recent QA models has been a problem in building a real-time QA system. This paper tries to solve the computational efficiency of recent MRC models. Their idea is somewhat similar to the idea used in phrase-indexed QA in open-domain QA. Thus, the topic of this paper is well-aligned with the recent approaches in the QA community.

Weakness:

1. The authors converted some QA datasets (ShARC and HybridQA) to long-document QA forms and evaluated DocHopper on these datasets. However, the authors did not show evidence that their modification on ShARC and HybridQA datasets is valid.
2. The runtime efficiency of DocHopper is one of the main contributions that the authors claim in their paper. However, this paper does not provide enough information on measuring the runtime. For example, does the runtimes of the baselines and DocHopper measured in a fair environment? Please provide more detailed settings used in measuring runtime efficiency.
3. IRRR is a multi-hop QA model in the full-wiki setting, and HotpotQA-long is in the distractor setting. Therefore, comparing IRRR and DocHopper is not a fair comparison. Although the authors provide the performance of HGN (in distractor setting), the performance of DocHopper underperforms. However, this is not a weak point since one of the research problems of this paper is alleviating the runtime problem while maintaining the MRC performance. However, since it is not clear how the authors conduct experiments to compute the runtime of each model, the HotpotQA-long results become insufficient evidence.

**Summary Of The Paper:**

This paper provides a novel MRC model (DocHopper) for multi-hop QA over long structured documents. In multi-hop QA, the evidence necessary to answer a user's question is spread across different parts of the long document. Previous approaches find the evidence by iteratively updating the user's query. The problems in these previous approaches are 1) computational efficiency and 2) ineffective modeling strategy for figuring out the relations of the evidence. DocHopper resolves these problems with a hierarchical attention mechanism. Hierarchical attention mechanism provides two types of embedding vectors for a single paragraph: 1) local sentence vectors and 2) global context vector of the paragraph. DocHopper computes the similarities between the query vectors and these sentence/paragraph vectors and selects the proper evidence. Since the sentence/paragraph vectors can be pre-computed, the only inference time required for this method is the time for question embedding, and this brings drastic improvement in the computational efficiency. This paper uses four types of datasets for evaluation: 1) conversational QA (ShARC), 2) TableQA (HybridQA), 3) QA on academic paper (QASPER), and 4) multi-hop factual QA (HotpotQA), and shows the QA performance and computational efficiency of their model.

**Summary Of The Review:**

The research problem of this paper is the inefficient computation time of existing QA models. This paper proposes a novel machine reading comprehension model for a long document. This paper provides evaluation results of their model on four types of datasets, and some experimental results show that DocHopper outperforms other baselines models. However, some evaluations results (results on ShARC and HybridQA) are insufficient to show the validity of their idea since these datasets are not designed for long-document QA. Also, it is unclear which experimental setting the authors used to compute the runtime of their model and baselines.

Since my concerns are from the unclear description of this paper, I'm willing to increase the overall recommendation score if the authors provide a more clear description.

---

> ### Author Response · Authors · 2021-11-22
> **Response to Reviewer XkRC**
>
> Thanks for your comments on the paper. We apologize if the current draft is unclear, but we **do not agree** that the main contribution of this paper is “*the inefficient computation time of existing QA models*”. We tackle the problem of answering questions with long context, especially when the questions are multi-hop. Few existing models and datasets have been proposed for this task. DocHopper is one of the first models that are designed for this problem. The computation efficiency is surprising but it is a plus.
>
> To answer your question:
> 1. We discussed the data preparation process in Appendix A.1. We quote the description for the ShARC dataset here: “*we consider instead a more challenging long-document setting, in which the snippet is not known, and the model must also locate the snippet from the document. We crawl the web pages with the provided URL…*” We do not modify the HybridQA dataset. We simply remove the cell boundary and convert them into plain text. Please refer to the Appendix for more details. For both datasets, we **did not change the answers or remove any context**.
>
> 2. We stated in the paper that “*runtime is measured by re-running their open-sourced code*” and that “*runtime is measured as examples per second with a batch size of 1*”. We ran the experiments with the Titan-X GPU with 12GB memory.
> (1) Comparison to HGN: The performance of DocHopper on HotpotQA-Long and HGN on HotpotQA (distractor) is **not comparable** as the context for each question in HotpotQA-Long is **11 times longer** than that in the distractor setting. Please note that reading longer context makes the task more challenging as it is harder for models to find relevant information. It is surprising that the performance of DocHopper is only 4 points lower.
> (2) Comparison to IRRR: Again, HotpotQA-Long is **different from the original distractor-setting** because it has much longer context. No existing reading comprehension models have been designed for tasks like this. IRRR is a “retrieve-and-read” pipeline that could be applied for this task that first retrieves relevant information and then reads them to find the answers.

---

### Official Review · Reviewer_qv3L · 2021-11-02

**Correctness:** 2
**Technical Novelty And Significance:** 2
**Empirical Novelty And Significance:** 2
**Recommendation:** 5
**Confidence:** 3

**Main Review:**

1. If it is claimed as "hierarchical", I would expect something like a "general and narrow down" strategy. Unfortunately, what the model actually does is just simply flatten every paragraph (see the definition of content embedding C and Eqs. 4, 5, and 6). The only "hierarchical" stuff is a simple add-on embedding that is a simple weighted sum of all sentences. Also, the "iterative" comes from the problem definition itself while having nothing to do with the method. In this sense, the proposed attention model is neither "iterative" nor "hierarchical". I do think it makes novel technical contributions.

2. The process when attending to a paragraph vector (i.e., unpack the paragraph by reusing dot-product attention, then pack it again with the extra learnable attention) seems tricky and cumbersome but the intuition behind is not clearly illustrated.

3. The experiments on the expanded datasets are unfair to other baselines. As their deeply contextual QA interaction either falls in token-level or is subject to the ability of the backbones (e.g., BERT’s 512 length limit). I would recommend authors provide results on the original dataset.

4. The performance of the proposed model lags far behind MATE on HybridQA dataset. It is curious that how MATE would be performed if applied to QASPER dataset.

Minor comment:

Strictly, the softmax term in Eq.3 (and argmax term q_t c_m) should be s^i_j q_t^T, as q_t and s^i_j are both 1*d matrices according to the notation at the bottom of page 3.




**Summary Of The Paper:**

The paper proposes a simple attention-based model for conversational and multi-hop QA tasks. The model use BERT-like pre-trained LM ETC separately encodes questions and paragraph (i.e., a collection of sentences). Besides the encodings on sentence-level, the final context encodings also contain extra paragraph embeddings, which are a weighted sum of sentences’ encodings using a simple dot product attention. For the QA interaction, the models use a hard attention mechanism to select an entry representing either a sentence or a paragraph.
The experiments on two extractive QA datasets HYBRIDQA and QASPER show the model performs worse than the MATE model on HYBRIDQA but marginally better than other baselines on QASPER. On multi-hop QA and conversational QA tasks, the model performs marginally better than baselines on *expanded dataset*, but authors do not provide results on original datasets.


**Summary Of The Review:**

Trivial method, contributing little scientific knowledge.

---

> ### Author Response · Authors · 2021-11-22
> **Response to Reviewer qv3L**
>
> Thanks for your review.
>
> 1. “Not hierarchical”: We are glad that the reviewer finds it surprising that “*a simple add-on embedding that is a simple weighted sum of all sentences*” is sufficient to capture hierarchical information at a higher level. Experiments show that, without this **simple** layer, the performance drops 3~10 points in different datasets (Table 1 and 3). We agree that more complicated model structures with more expensive computation could possibly work better, but we would like to save it for future work.
>
> 2. “Not iterative”: We perform **iterative attention to collect and aggregate evidence** (Eq. 4 and 6) and **iteratively update the queries** (Eq. 7). Experiments in Table 1 and 3 show that the performance drops 3~20 points without the iterative process (“single-hop”).
>
> 3. “Unfair baselines”: We thank the reviewers for pointing out the weakness of existing baselines, e.g. BERT with 512 length limit. Most recent reading comprehension models  focus on understanding short contents and existing datasets are specifically simplified due to the limit of the models. To the best of our knowledge, very few existing reading comprehension models have been designed to read long documents (**with tens of thousands of tokens**), and as of now no more datasets (besides QASPER) are available.
>
> 4. “MATE on QASPER”: In the “Result and Analysis” part in Section 4.1, we extensively discuss the difference between MATE and DocHopper. To quote it here: “*MATE is specifically designed and pretrained to understand tabular data, while DOCHOPPER is applied to general documents*”. So MATE cannot be applied to any other datasets experimented in this paper.

---

> > ### Comment · Reviewer_qv3L · 2021-11-29
> > **Reply authors' response**
> >
> > Thanks for the efforts made in the rebuttal period. The response addressed part of my concerns, esp. Q3 and Q4. For "hierarchical", I agree with reviewer zXR3 that it is better to rephrase this terminology to avoid confusion with the multi-layer model. I understand that authors aim at a simple yet effective layer for multi-hop QA on long documents, however, its general effectiveness is not clear due to the insufficient experimental settings (also pointed out by other reviewers) and the scarcity of evaluating datasets. Thus, I keep my initial comments that the technical novelty is limited. Will change my rating accordingly.

---

### Official Review · Reviewer_zXR3 · 2021-11-02

**Correctness:** 2
**Technical Novelty And Significance:** 2
**Empirical Novelty And Significance:** 2
**Recommendation:** 5
**Confidence:** 4

**Main Review:**

The proposed approach is interesting and more efficient compared with cross encoding approaches at inference time.
Most prior results in the literature suggest that separate encoding approaches work inferior to cross encoders, thus results in this paper are encouraging. Despite encouraging results, there are some shortcomings in the execution.

- One critical result from last year's research was that Transformers can capture multihop relationships in long documents without the need for directly modeling multi-hop behavior and hierarchical structures (see BigBird, ETC, Longformer papers). This paper somewhat goes against that direction and suggests that explicitly encoding multi-hop information can improve results. However, it remains difficult to fully accept this claim as some of the key comparisons are not fully controlled. More details below.

- As mentioned in the paper, the ETC baseline is hampered by the 4096 context size limit. For the baseline, the authors encode the context paragraph by paragraph to sidestep the issue. However, this makes this baseline significantly weaker. Why not just evaluate the proposed model on the original HotpotQA data so that results are directly comparable? In addition Longformer can handle sequences of up to 16K tokens. I would have liked to see a Longformer based encoder as retriever + a BERT based reader as another more fair baseline on all the datasets.

- While motivation for longer documents is important, most of the datasets used are not the original versions. E.g., Hotpot and ShARC are modified to make them longer and for Qasper only a subset of extractive answers are used. This makes direct comparisons with SOTA not possible.

- I would have liked to see a comparison of number of parameters of the proposed model compared with baselines. For example, the LED baseline for QASPER is a single stage model where as the proposed approach uses two models, one for extracting evidence + a BERT-based reader for extracting the final answer. In this regard, the comparison is not entirely fair.

- The exact method of constructing the paragraph embeddings by a weighted average of sentences based on query relevance is a bit under motivated. It would have been nice to see an ablation on importance of this design (e.g., compared with simple or learned weighted average pooling).

- One of the stated benefits of the proposed approach is that it is more efficient by reducing the need for cross encoding the query and context at inference time. However, I would have liked to see some discussion about the added storage cost compared with the cross encoding methods. Pre-computing and storing vectors of all sentences and paragraphs in a document can be incur significant storage costs.

- It would have been nice to see some error analysis, case study, or some discussion looking at what types of information the proposed model is able to capture compared with the baselines.

Questions for authors:
- What is finally used as text of q_null?
- Which ETC model size is used? Which model size is used for the BERT-based reader?

Terminology:
- I'm not sure "hierarchical attention" is standard terminology for the model architecture of ETC. It might be confused by multi-layer models that are designed so that lower layers encode smaller pieces of information (e.g, words) and subsequent layers ingest these fine-grained representations and produce coarser level representations (Yang et al 2016). I would either use another terminology or add an explicit definition to make it clear that by "hierarchical" you mean that the model provides representation of individual blocks of text within the full sequence, as ETC does not have an explicit hierarchical structure.

**Summary Of The Paper:**

The paper proposes an iterative approach for multi-hop question answering. At high-level the proposed model breaks a question into multiple sub-questions and then adds information relevant to each sub-question to the query vector for the next step retrieval. At each iteration an ETC encoder is used to encode the document and a sub-question; the vector corresponding to sub-question is then updated iteratively by contextualizing it over sentences and paragraphs in the document and is used to extract the final answer using a subsequent BERT reader. Evaluation results show improvements on 3 of 4 datasets.

**Summary Of The Review:**

Overall, the paper proposes an interesting approach to answer questions over long documents efficiently. However, there are some issues with the execution, presented results, and analyses making it difficult to fully accept the main claims in the paper.

---

> ### Author Response · Authors · 2021-11-22
> **Response to Reviewer zXR3**
>
> Thanks for your valuable feedback. The reviewer’s major concerns are (1) not using original datasets, and (2) comparison to ETC. Please see our response below with new (or existing) experiment results.
>
> 1. “Comparison to ETC / Longformer.” We realize ETC’s strong performance on multi-hop reasoning and build an additional module on top to improve its reasoning ability. While ETC works well on 2-hop factual questions, e.g. in HotpotQA, previous experiments have not shown its capability on more than 2-hop questions, e.g. ShARC. In ShARC, the model should follow the multi-turn conversation and answer the question at the end. In this task, the performance of ETC on ShARC is only **61.1** compared to DocHopper **72.4**. On QASPER, models need to aggregate multiple information at different levels. The performance of the Longformer-based model is **26.1** F1 compared to DocHopper **29.6**.
>
> 2. “Datasets are not original.” We find it unfortunate but QASPER is the only dataset that is close to the long-document reading comprehension task. Most of the existing datasets, e.g. HotpotQA, ShARC, etc. are designed for paragraph-level reading comprehension. While such tasks are interesting, they are **deliberately made easy** as previous machine reading models are not capable of handling long context. For example in HotpotQA, it only includes the first paragraph of Wikipedia articles. This is very unnatural because in real scenarios evidence and answers could appear anywhere in the documents. We emphasize the need of reading longer documents, and we modify existing datasets to test the model’s performance.
>
> 3. “Results on QASPER and modified HotpotQA.” DocHopper works reasonably well in the long document setting. Even though the context of HotpotQA is **11 times longer**, the performance of DocHopper is only **4 points lower** than in the original setting. We are training a downstream generative model for the remaining abstractive questions in QASPER. We will add the numbers later.
>
> 4. “Longformer retriever + BERT reader” We run the experiments suggested by the reviewer that uses Longformer as a retriever and uses BERT to read the retrieved sentences. On QASPER, the performance of the proposed pipeline is **12.8 / 26.5** (EM / F1) compared to DocHopper **14.0 / 29.6**. On HotpotQA-Long, its performance is **50.4 / 61.2** compared to DocHopper **57.4 / 69.5**. It shows Longformer is not a good “retriever” from long context for multi-hop questions.
>
> 5. “Number of parameters” DocHopper finetunes ETC-base with 166M parameters. The additional layers are all linear and have <1M parameters in total. The BERT module contains 110M parameters. In total DocHopper has 277M parameters. LED in QASPER has 154M parameters. DocHopper needs an additional BERT reader because DocHopper does not jointly encode questions and context. Even if we pair Longformer with a BERT reader, the performance is still >2 points lower than DocHopper.
>
> 6. “Storage” DocHopper only stores one embedding per sentence (not per subword token). It only takes **0.65Mb** to store the precomputed context embeddings. We claim that this storage usage is relatively cheap.
>
> 7. “Constructing paragraph embeddings” We run the proposed ablated experiment with simple mean pooling and learned weight pooling. With simple mean pooling, the performance on HybridQA drops from **47.7 to 46.3**. With a separately learned pooling layer, the performance on HybridQA drops from **47.7 to 46.8**. The proposed query-dependent weighting is a simple and efficient way of computing a weighted summed embedding. A better designed paragraph embedding layer may further improve the overall performance but we will leave it for future work.
>
> Quick answer:
> 1. q_null is not derived from text. It is a variable that will be learned.
> 2. We use ETC base and BERT base.

---

### Official Review · Reviewer_Ld3M · 2021-11-03

**Correctness:** 2
**Technical Novelty And Significance:** 2
**Empirical Novelty And Significance:** 2
**Recommendation:** 5
**Confidence:** 3

**Main Review:**

### Strengths
I think overall the proposed approache is technically sound well motivated by the challenges in complex reasoning over a long context. They also test their approaches on diverse datasets including conversational QA and show the applicability to diverse QA data.
Their proposed method largely outperforms its base model, ETC, showing the effectiveness of newly introduced modules. Also, removing the necessity of jointly encoding query-document and reusing the encoded document embedding is useful to improve the run time efficiency.

### Weaknesses
I have two main concerns listed below.
1. **Limited novelties and technical contributions**: although the method seems to be technically sound, the main component is based on ETC, and from my understanding, their technical contribution mainly lies in iterative attention to update the query representations in embedding space. Updating query representations in embedding spaces instead of actually updating query text for multi-step retrieval&reasoning have been studied in prior work such as Feldman and El-Yaniv (2019) or Das et al. (2019), but those papers are not cited, and I would like to see comparison or discussions on the differences between those studies.

2. **Comparisons with the prior work**: Although they claim the proposed approach achieves state-of-the-art results on three of the benchmarks, the experimental settings are often slightly different from the original settings (e.g., only evaluate on the subset of the questions, use more passages than the default settings such as HotpotQA-Long). Claiming state-of-the-art results on those variant settings may not be fair, especially when the baselines to be compared are not specifically designed for the new setting.

In addition, I sometimes have difficulties following the paper due to some missing details in the method section. For example, I think there should be more details on how "attention is supervised" in the loss function paragraph in Section 3.4, instead of talking about them in the implementation details. On the other hand, how to embed queries for each task is discussed in detail in Section 3.2 before talking about DocHopper's technical contribution (Section 3.4. Iterative attention). This makes it hard for me to understand what is the key technical contribution of DocHopper, and which parts are generally appreciable to different tasks, and which parts should be customized for each task.

### Missing reference
- Feldman and El-Yaniv. 2020. Multi-Hop Paragraph Retrieval for Open-Domain Question Answering. In Proc. ACL.
- Das et al. 2019. Multi-step retriever-reader interaction for scalable open-domain question answering. In Proc. ICLR.

**Summary Of The Paper:**

This paper introduces DocHopper, a new model for complex question answering over long documents (e.g., multi-hop QA over multiple paragraphs, conversational QA, reasoning over scientific documents). DocHopper is based on ETC (Ainslie et al., 2020) and DocHopper extends the existing hierarchical attentions from ETC with a new approach to update query representations in latent space. Their model does not jointly encode a question and context and does not require re-encoding of queries as in prior work, which leads to their effectiveness at inference time. They evaluate DocHopper on four different datasets: ShARC, QASPER, HotpotQA, and HybridQA. The proposed method achieves strong performance on those datasets, reducing the computational cost at the inference time.

**Summary Of The Review:**

In summary, I think the proposed approach is technically sound and a nice extension of ETC, but I don't agree that it provides significant technical contributions. Experimental results are strong, but as the experimental settings are different from the original setting, I do not think the paper should claim DOCHOPPER achieves state-of-the-art results on three datasets.

---

> ### Author Response · Authors · 2021-11-22
> **Response to Reviewer Ld3M**
>
> Thanks for your comments.
>
> 1. We understand your concern about the modified datasets. Yes, experimenting with modified datasets is definitely not ideal. But unfortunately, QASPER is the only dataset that is close to the long-document reading comprehension task. Most of the existing datasets, e.g. HotpotQA, ShARC, etc. are designed for paragraph-level reading comprehension. While such tasks are interesting, they are **deliberately made easy** as previous machine reading models are not capable of handling long context. For example in HotpotQA, it only includes the first paragraph of Wikipedia articles. This is very unnatural because in real scenarios evidence and answers could appear anywhere in the documents. We emphasize the need of reading longer documents, and we modify existing datasets to test the model’s performance. We are training a downstream generative model for the remaining abstractive questions in QASPER. We will add the numbers later.
>
> 2. We thank the reviewer for pointing out the missing citations. We will discuss them in the related work. The papers are relevant in the sense of updating the queries in the embedding space, but they focus on the open-domain QA setting. Here, we focus on a **different long-document setting**, where models read a long document that contains coherent information on a certain topic. Answering complex, multi-hop questions over long documents requires navigating through different parts of the documents to find different pieces of information. This makes sequentially retrieving one passage at a time sub-optimal. DocHopper is able to look at information at both coarse and fine levels to predict the final answers.

---

### Decision · Program_Chairs · 2022-01-20

**Decision:**

Reject

**Comment:**

Strength
* The paper is relatively clearly written.
* The proposed method appears to be sound.

Weakness
* The novelty of the work seems to be limited.
* The experiment part needs significant improvements.  The comparison with existing methods may not be fair.  Evaluation of efficiency should be given. There are also detailed investigations that need to be conducted, as indicated by the reviewers.
* There are technical issues that need to be addressed.